# The Impact of COVID-19 on the Service of Emergency Department

**DOI:** 10.3390/healthcare9101295

**Published:** 2021-09-29

**Authors:** Shaia Alharthi, Modi Al-Moteri, Virginia Plummer, Abdulellah Al Thobiaty

**Affiliations:** 1Nursing Department, College of Applied Medical Sciences, Taif University, P.O. Box 11099, Taif 21944, Saudi Arabia; shaiaa@moh.gov.sa (S.A.); althobaity@hotmail.com (A.A.T.); 2School of Health, Federation University, Mount Helen, VIC 3350, Australia; v.plummer@federation.edu.au; 3School of Nursing and Midwifery, Monash University, Frankston, VIC 319, Australia; 4Peninsula Health, Frankston, VIC 3199, Australia

**Keywords:** emergency department, COVID-19, triaging

## Abstract

(1) Introduction: the COVID-19 pandemic significantly impacted the number and acuity of emergency departments (ED) patients, specifically those with non-COVID-19-related health problems. However, the exact impact of the COVID-19 pandemic on ED services is the subject of comprehensive debate. (2) Aim: to gain insight into the consequences of the first wave of the COVID-19 pandemic based on non-COVID-19 presentations and patient acuity using the Canadian Triage and Acuity Scale (CTAS). (3) Method: in Phase 1, the ED records of one of the main regional non-COVID-19 hospitals in Saudi Arabia were retrospectively audited from August 2020 to February 2021—after the first wave of COVID-19—then compared to information collected for the same period in previous year. Phase 2 included calculating the waiting time to identify delays and issues that may impact the triage effectiveness. (4) Results: a change across all CTAS levels was observed post the 1st wave of COVID-19 pandemic. Specifically, there was an increase in the number of patients presenting as higher acuity (CTAS 1 and 2) and a decrease in patients presenting as lower acuity (CTAS 4 and 5). Longer waiting times for patients presenting to ED were also reported**.** Specifically, 83% of patients presenting as higher acuity experienced a delay. (5) Conclusion: further studies are required to investigate association between the 1st wave of COVID-19 and patient presentations and/or acuity or patient demand and ED capacity.

## 1. Introduction

Patients arriving for emergency care go through a specific process in which their health problems are assessed and categorized into certain categories according to their urgency (1–5). This screening and categorization process is referred to as ‘triage process’ [1]. The triage process has a positive control in number and acuity of patients presenting to Emergency Departments (EDs) [2] through supporting the clinicians’ decision-making, consequently decreasing the occurrence of patients’ adverse outcomes [3]. However, the crisis of COVID-19 pandemic significantly impacted the number and acuity of ED patients, specifically those with non-COVID-19 related health problems [4]. Indeed, during 2020, a decline of around 25–50% was observed in patients visiting EDs and presenting with cardiac symptoms compared with that of 2019 [4].

COVID-19 is a highly infectious respiratory disease and it spread rapidly around the globe since 2019. The World Health Organization (WHO) declared COVID-19 as a pandemic and a global concern [5]. In the Kingdom of Saudi Arabia (KSA), the first case was confirmed and declared in March 2020. The number of confirmed cases reached the peak in May and June 2020 and fluctuated during July of that year. However, from August 2020 the number of confirmed cases decreased dramatically and reached the minimum in February 2021 [6].

As elsewhere, in KSA, during the peak times of COVID-19, the number of patients presenting to ED with COVID-19 increased causing alteration in the number of visits of non-COVID-19-related conditions and perhaps a delay in timely care seeking. This might be concluded by the growing body of global knowledge showing that one year after the occurrence of the pandemic, there was a worldwide change in the numbers of non-COVID-19 patients with both critical and noncritical conditions who present to EDs [7,8,9,10]. For instance, many reports at national and international level showed that survivors of car accidents, people presenting with symptoms of possible origin, cardiac, back, or limb pain and infectious diseases significantly decreased during COVID-19 pandemic [4,7,8,9,10]. This might have critical consequences on patient mortality and morbidity. Authors on these studies stressed the need for having well-established strategies during pandemics to ensure patients are seeking and receiving timely medical attention.

In general, the impact of the COVID-19 pandemic on the “triage process” in ED is the subject of comprehensive debate [11,12]. The impact of the first wave of COVID-19 pandemic on ED presentations and acuity in KSA is unknown. This study attempts to fill this gap by retrospectively auditing hospital ED admission records of a non-COVID-19 hospital in KSA and comparing the number of patients presenting with lower and higher acuity before and after the COVID-19 pandemic. This study sought to identify trends and gain insights into the consequences of the first wave of the COVID-19 pandemic on non-COVID-19 patient acuity, allowing healthcare policy makers to better understand this phenomenon and prepare for future pandemics.

## 2. Materials and Methods

Quantitative descriptive research design was conducted. An audit of hospital ED admission records to review the impact of the COVID-19 pandemic on the triage process in the ED of a main regional non-COVID-19 hospital in KSA.

### 2.1. Study Setting

Ministry of Health (MOH) in KSA at the beginning of the first wave of the COVID-19 pandemic declared some of its hospitals to be completely “non-COVID-19” hospitals. The study was conducted at one of these hospitals. The hospital ED encompasses 30 beds as follows, 2 beds (Category 1), 9 beds (Category 2), 13 beds (Category 3), and 6 beds to serve cold cases (Category 4, 5) according to their urgency. The staff of the ED assesses and treats approximately 110,000 adult patients per year. There are three shifts for the physicians and nurses: morning (7 a.m.–3 p.m.), evening (3 p.m.–11 p.m.), and night (11 p.m.–7 a.m.). Physicians in total were 1 ED consultant, 4 ED specialists, and 12 residents covering the emergency rooms over the 24-h period. Meanwhile, each shift is covered by two nurses in (Category 1), three nurses (Category 2), five nurses (Category 3), five nurses for (Category 4, 5 and the triage point). At peak times, and usually during weekends, there are an additional two nurses covering the visual triage area.

### 2.2. Data Collection Tool

In Phase 1, the records of the ED were retrospectively examined to determine the number of patients who were triaged in the six-month period from August 2020 to February 2021—after the first wave of COVID-19—and were then compared to that of information collected for the same period in previous year (August 2019 to February 2020). Each patient presenting to EDs was evaluated and assigned a triage category based on the Canadian Triage and Acuity Scale (CTAS) by a registered nurse (RN). Based on triage acuity level, the patient will be assessed by an emergency physician. CTAS is “Quick Look” 5-level tool to determine the patient’s triage level [3]. It was used in KSA hospitals for the past 15 years [13]. The tool is based on patient complaints, and each complaint has a comprehensive description and covers high-risk markers as follows: 1st level “Resuscitation: to be seen immediately”; 2nd level “Emergent: to be seen <15 min”; 3rd level “Urgent: to be seen <30 min”; 4th level “Less Urgent: to be seen <60 min”; and 5th level “Nonurgent: to be seen <120 min” [14].

Phase 2 was conducted in March 2021 and included recording data over two weeks by observing patients going through the triage process. The observations were conducted using a predesigned checklist Appendix A). The information collected about the patients in the study included age, gender, and residence. Two processes contributed to calculating the waiting time, namely, before triaging and after triaging [15]. Before triaging is the time from arriving to processing the patient into a triage category, and is mainly carried out by a RN. After triaging is the time from triaging to the physician’s examination to determine the appropriate treatment. Before and after triaging, waiting time was calculated to identify delays and issues that may impact the triage process effectiveness. The time beyond the recommended time of the CTAS was considered delayed [14].

### 2.3. Data Analysis

The data were entered and analyzed using Statistical Package for the Social Science (SPSS) Version 22. Sample characteristics were examined using descriptive analysis. The total number of CTAS cases were tabulated for each category and month, and then summed for the period of August 2019 to February 2020 and of August 2020 to February 2021. Chi-square tests of independence were then conducted to compare the proportion of CTAS categories in 2019–2020 to CTAS categories in 2020–2021. Qualitative data was organized, and similar segments were grouped and categorized into a set of concepts. Similar concepts were then grouped into broad codes (Factors). Content analysis was conducted to quantify the occurrence of these codes (Factors) in the set of data.

## 3. Results

The total number of ED visits by the public who were triaged from August 2020 to February 2021 at the end of the first wave of COVID-19 was (*n* = 94738), while the number of ED visits by the public who were triaged during the same period of time in 2019 and 2020 was (*n* = 97941). The Chi Square Test of Independence was used to calculate the difference between the two sets of data. Overall, reviewing ED database revealed that the number of visits during 2020–2021 was 3% less than that of 2019–2020, with a noticeable decline in patients presenting as lower acuity (CTAS 4 and 5). There was also an increase (>200%) in the number of patients presenting as higher acuity (CTAS 1 and2) (Table 1).

There were 2185 ED visits during March 2021 (2nd Phase). Out of the 2185 visits, 713 participants (33%) were included in the study. Of these patients, 647 emergency patients were from Levels 1–3, and 66 patients were from Levels 4–5. Descriptive analysis in the form of frequency was used to describe the characteristics of the sample. Patient demographic data and the shift of their visit are shown in Table 2. Almost three quarters of patients were male (*n* = 528, 74%), while (*n* = 189, 26%) were female. There was almost equal distribution of age groups of 18 to 40 years (*n* = 268, 38 %), 40 to 59 years (*n* = 256, 36 %), and >59 years (*n* = 189, 26 %), and equal distribution of the time of the shift of visit: morning (*n* = 361, 37 %), afternoon (*n* = 251, 35 %), and night (*n* = 201, 28 %). Most of the patients lived in the areas served by the hospital (*n* = 631, 88.5%).

Out of the 713 patients who visited the ED and included in the study, 335 (47%) experienced a delay. The delay was operationally defined as ‘a delay in the time recommended by CTAS guidelines for a physician to assess patient and initiate treatment based on triage acuity level’. Delays were evidenced in pre- and post-triaging as shown in Table 3. It was found that the total average delay time for patients from arriving to physicians’ examination was around 15, 35, and 46 min for patients of Levels 1, 2, and 3, respectively. The most common reason for the delays was ‘crowdedness’ (88%) as shown in Table 4. This can be attributed to increase numbers of critical patients causing a mismatch between patients’ demand and the ED’s capacity to deliver timely care.

## 4. Discussion

The study findings revealed that there was a slight decline in the number of patients visiting the ED at the end of the first wave of COVID-19 compared with that of the same period in the previous year at the regional hospital in KSA. Further, there was a noticeable change in patients’ acuity levels. These findings are consistent with the finding of a study conducted by Ferron et al. (2021) and Boserup et al. (2020) [16,17] who reported an increase in the number of patients presenting as high acuity (CTAS Level 1) at the end of the first wave of COVID-19.

Though the hospital included in this study was a non-COVID hospital, an increase in the number of patients with higher acuity of illness visiting ED and a decrease in the number of patients presenting as lower acuity (CTAS level 4 and 5) after the end of COVID-19 first wave possibly resulted from several factors which contributed to such changes. The instructions issued to people by the Saudi government concerning social restrictions might be one of the reasons for the decrease in EDs visits. Further, during the state of COVID-19 pandemic, people’s reluctance to go outside their homes is likely to be associated with fear of catching COVID-19 infection [17]. However, a critical consequence of these changes is that patients with high acuity illness or injury may wait too long to be treated adequately. It is important to increase people’s awareness that potentially serious conditions require medical or surgical attention during COVID-19 with the same urgency as they did prior to the pandemic.

Such inconsistent changes across the CTAS five levels implies that COVID-19 may lead to actual changes in ED services, and it will be of interest to consider this issue for future pandemics. Preparation for a surge of potential patients with the highest acuity of illness is required [18].

Despite the decrease in the number of patients presenting as lower acuity (CTAS level 4 and 5), there was a delayed waiting time beyond the time recommended by the *CTAS.* Waiting time was long in the current study. An important factor that must be considered when calculating the waiting time is the baseline of the waiting time of the EDs in the hospital of the study. However, data supporting calculating the waiting time for the same period in the previous year were not available. Hence, this study was an “opportunistic observational study”, and results cannot be compared with that of the previous year. A “repeated measures scientific study” is required.

The current study may indicate that the influence of the COVID-19 pandemic overwhelms the patient acuity and triaging time. Studies showed that measures taken during COVID-19 emergency led to a significant drop in emergency services, and soon the EDs will encounter an increase in the number of patients with worse prognosis seeking urgent medical attention and overcrowded emergency services [19]. Increasing the number of people with serious health problems after the pandemic leads to a mismatch in ED supplies and demands. Crowdedness can reduce the quality of the care delivered and may have a negative impact on patient outcomes [19]. There are clear indications that ED services require careful planning and management of resources pre- and during pandemic [4].

## 5. Implications

The study findings indicate that COVID-19 pandemic possibly had unintended consequences on ED services. It is possible that patients are waiting at home too long during the pandemic before seeking care in EDs and they arrive in a more severe state arising from postponed care. Such delays possibly led to an increase in the number of patients visiting EDs and presenting with serious health conditions. Knowing these patterns of presentations allows decision makers to predict and plan for future pandemics. Future studies are required to investigate apparent reluctance to present to ED in the early phase of particular conditions and health outcomes and what public health messages are likely to be impactful during the COVID-19 pandemic.

## 6. Limitations

This study was conducted in the ED of a specific geographical area of KSA, and it may not apply to other geographical areas or countries. Different healthcare institutions may demonstrate different trends. The study was conducted for 6 months after the first wave of COVID-19. The results may change after a longer-term follow-up. In addition, Phase 2 in the study was limited to specific time. Hence, the results in Phase 2 can be influenced by the patients and staff in period of the study. Comparing delay time of patients in EDs after Phase 1 of COVID-19 with the same period in previous year was not investigated. More studies to cover the whole year are required.

## 7. Conclusions

There was a decrease in the number of patients presenting to ED as lower acuity after the COVID-19 pandemic, and meanwhile, there was an increase in the number of cases presenting to ED as higher acuity. It is essential to determine whether these changes are associated with a mismatch in long-term between patients’ demand and the ED’s capacity to deliver timely care. If confirmed, this requires careful planning and management of resources pre- and during pandemic. The study findings may be of interest to consider for future pandemics, including the prediction of efforts required to adjust ED triaging services.

## Figures and Tables

**Table 1 healthcare-09-01295-t001:** Comparing CTAS data base.

Half Year	CTAS1	CTAS2	CTAS3	CTAS4	CTAS5	Total
Aug 2019 to Feb 2020	588	8101	52,415	26,798	10,039	97,941
Aug 2020 to Feb 2021	1730	8709	63,408	18,202	2689	94,738
Change	1.94	0.08	0.21	−0.32	−0.73	−0.03
*p* value	<0.001	<0.001	<0.001	<0.001	<0.001	<0.001

**Table 2 healthcare-09-01295-t002:** Patient and assessor demographic data.

Item	Frequency	Percent
Age		
Young adult (18–39)	268	37.6
Middle aged adults (40–59)	256	36
Older adults (>59)	189	26.4
Gender		
Male	528	74.1
Female	185	25.9
Residence in areas served by the hospital		
Yes	631	88.5
No	82	11.5
Shift		
Morning	361	36.6
Afternoon	251	35.2
Night	201	28.2
Assessor work experience		
<5	378	53.0
5 to 10	289	40.5
10>	46	6.5

**Table 3 healthcare-09-01295-t003:** Average delay time.

	Number of Cases	Delayed Cases (%)	Before Triaging	After Triaging	Total Average Delay Time
CTAS 1	34	34 (100)	0:8:37	0:5:10	0:13:47
CTAS 2	85	65 (76.4)	0:20:08	0:14:50	0:34:58
CTAS 3	528	202 (38.2)	0:15:11	0:31:10	0:46:21
CTAS 4	62	33 (53.2)	0:45:37	1:12:14	1:57:51
CTAS 5	4	1 (25.0)	2:23:00	1:17:00	3:40:00
Total	713	335 (47)			

**Table 4 healthcare-09-01295-t004:** Reasons for delay in ED.

Reason	N (%)
Crowdedness	298 (88%)
Long nursing handover time	13 (4%)
Getting busy with suddenly deteriorated patient	11 (3%)
File making process	8 (2%)
Multiple consultations with further investigation required	2 (0.6%)
Poor patient response to initial ED management	1 (0.3%)
Total	335 (100%)

## Data Availability

Not applicable.

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
