# Peer review of "The Impact of COVID-19 on the Service of Emergency Department"

_healthcare, 2021, doi:10.3390/healthcare9101295_

Round 1

Reviewer 1 Report

Thank you for this interesting manuscript. I’m sure clinicians and hospital administrators around the world are curious about this question, and will be interested to see how it worked in at least this studied hospital. Overall, the study is straightforward with interesting findings. However, some things need clarification, and there seems to be missing parts that the authors did perform but forgot to include in the manuscript’s methods/results/discussion (see below).

Abstract: When you state a change across all CTAS levels was observed, it would be nice to have something more specific.

Line 36: Can you be more specific as to what Butt et al found? What general ways was the number and acuity of ED patients affected by the pandemic?

Line 49: Again, can you be more specific on what these changes were? Should state more than just that there were changes.

Line 114: Very interesting general findings, and very clearly stated.

Line 118: I am confused what the manuscript defines as ED visits (n=94,000+) vs ED cases (n=2185) vs study participants (n=713), can you please clarify these? Do you mean these were the cases/patients that were samples for the study?

Line 129: How does this compare to the year before? The following table does not mean much unless compared with the year before as not sure if the composition for delays is the hospital’s baseline. Also, you have multiple table 1’s, please correct this.

Line 135: Was not clear in the methods how the reason for delay was captured, please clarify in the methods.

Line 140: Again, this data does not mean much unless compared to the previous year’s data.

Line 161-162: Cannot yet state that there was increased delay times because the previous year’s delay metrics have not yet been presented for comparison.

Line 175: One structural issue with this manuscript is that this was a “non-COVID” hospital, so theoretically why would individuals be reluctant to go to this hospital even though technically there were no COVID patients? Please address this in your discussion.

Author Response

Please see the attachment."

Reviewer 2 Report

Thank you for the opportunity to see the manuscript. The authors take up a very important issue that is taking place all over the world. Poor preparation and organization of ED work in many cases cause significant delays and a lack of adequate assistance for urgent patients. However, the article has many errors and shortcomings that need to be corrected:

1) The title is confusing. It is enough to simplify it and point out that it is about the impact of the COVID-19 pandemic on the service time of ED patients.

2) Only objectives are presented in the executive summary. The reader would like to read briefly in the summary all elements of the study (introduction, purpose, methodology, results, conclusions)

3) The authors indicated the source of data of patients who underwent triage. But how was information about the causes of delays in providing assistance obtained?

4) The work methodology assumes extensive statistical analysis. Where is she at work? Have the authors forgotten this key element of the manuscript? Without statistical analysis, all conclusions described without scientific evidence.

5) The literature is up-to-date but very sparse. It is mandatory to supplement them with global research on the impact of the COVID-19 pandemic on critical care and the role of ED. Here are examples to be included in the literature:

a) Slagman, Anna, et al. "Medical emergencies during the COVID-19 pandemic: An analysis of emergency department data in Germany." Deutsches Ärzteblatt International 117.33-34 (2020): 545.

b) Mitura K. "The impact of COVID-19 pandemic on critical care and surgical services availability." Critical Care Innovations 3.2 (2020): 43-50.

c) Boserup, B., McKenney, M., & Elkbuli, A. "The impact of the COVID-19 pandemic on emergency department visits and patient safety in the United States." The American journal of emergency medicine, 38.9 (2020): 1732-1736.

d) Sosnowska-Mlak O, Curt N, Pinet-Peralta LM. "Survival in sudden cardiac arrest in emergency room: case-control study." Critical Care Innovations 2.3 (2019): 1-10.

Author Response

Please see the attachment."

Round 2

Reviewer 1 Report

Addressed my concerns, or adequately stated the limitations of their article where changes could not be made. Thank you 

Author Response

thank you

Reviewer 2 Report

The results do not include statistical methods. Table 1 shows the statistical significance of p, but does not indicate which statistical method was used (Anova, T-test ??).
